# Provable Convergence of Clipped Normalized-gradient Heavy-Ball Momentum for Adversarial Attacks

## Abstract

Gradient-based adversarial attack is dominated by the sign-like regime. Specifically, the sign-momentum MI-FGSM, which is a variant of Polyak's heavy-ball in normalizing each gradient by its $L_1$-norm, has achieved remarkable empirical success. However, the sign operation inevitably loses information about the magnitude as well as the direction of gradient or momentum, leading to non-convergence even in simple convex cases. Gradient clipping is an effective rescaling technique in optimization, and its potential has recently been demonstrated in accelerating and stabilizing the training process for deep learning. In this paper, to circumvent the drawbacks of sign-like gradient-based attacks, we present a clipped momentum method, in which the normalized-gradient heavy-ball momentum (NGM) is clipped as the update direction. By using a new radius-varying clipping rule, the clipped NGM is proved to attain optimal averaging convergence for general constrained convex problems. The experiments demonstrate that it remarkably improves the performance of sign-like methods and then verify that the clipping technique can serve as an alternative to the sign operation in adversarial attacks.

## 1 Introduction

Generating adversarial examples for a given learning model can be formulated as a constrained optimization problem, in which the objective is to find a constrained perturbation for the input data that maximizes the model's loss function. To solve such an optimization problem, many gradient-based methods have been proposed. Typical state-of-the-art algorithms include FGSM (Goodfellow et al., 2015), I-FGSM (or Basic Iterative Method (BIM)) (Kurakin et al., 2017), PGD (Madry et al., 2018), MI-FGSM (Dong et al., 2018) and NI-FGSM (Lin et al., 2020).

FGSM, I-FGSM and PGD are directly established upon the gradient descent principle. In contrast to the regular formulation, each of them uses the sign of the gradient vector as its update direction. In the stochastic setting, such a kind of method is usually referred to as signSGD (Bernstein et al., 2018). Similar to signSGD, MI-FGSM (Dong et al., 2018) uses the sign of a variant of Polyak's heavy-ball (HB) momentum (Polyak, 1964) as its iterative direction, in which each past gradient is normalized by its $L_1$-norm. By accumulating the past gradients in such a normalized way, MI-FGSM can stabilize update directions and then remarkably boost the transferability of adversarial examples. With MI-FGSM, they won the first place in NIPS 2017 Non-targeted Adversarial Attack and Targeted Adversarial Attack competitions (Dong et al., 2018).

So far, the gradient-based adversarial attack has been dominated by the sign-like regime due to its simplicity and effectiveness. SignSGD is popularly interesting since it can rescale and compress the gradient to alleviate the communication bottleneck in distributed optimization. For I-FGSM and MI-FGSM, their sign operation enables the iteration to lie within the constrained domain by simply restricting a small step-size. The convergence of sign-like methods has been established for smooth functions (Bernstein et al., 2018). Despite its empirical and theoretical success, simple convex counter-examples show that signSGD does not converge to the optimum. Even when it converges, signSGD may generalize poorly in comparison with SGD (Karimireddy et al., 2019). These issues arise because the sign operation sacrifices information about the magnitude as well as the direction of gradient (Karimireddy et al., 2019). This may explain why I-FGSM usually leads to instability and

inefficiency in adversarial optimization and why MI-FGSM and NI-FGSM are rarely used in pure optimization. To circumvent these drawbacks, in this paper, we will employ the clipping technique to deal with the iterative direction instead of the sign operation.

Gradient clipping has been long recognized as a rescaling technique in the development of gradient methods, and it has been proved to be an effective technique for optimizing functions with rapid growth (Alber et al., 1998). In contrast to the sign operation, it shrinks an individual gradient only when its magnitude exceeds a predefined threshold while keeping the direction unchanged. Intuitively, by ensuring that the gradient magnitude is not too large, the convergence of iterates becomes more well-behaved (Zhang et al., 2020b). Such an intuition has been made precisely in many recent works (Menon et al., 2020). More recently, theoretical analysis reveals that the clipping technique not only accelerates and stabilizes the optimization process but also dramatically improves the convergence properties of SGD without adding any additional costs to the original update procedure (Zhang et al., 2020b;a; Mai & Johansson, 2021). For training deep neural networks, it is efficient in relieving the exploding gradient problem from empirical studies (Pascanu et al., 2013). Extensive experiments on a variety of different learning tasks have illustrated that the clipping-like algorithms consistently perform better than the vanilla ones (Zhang et al., 2020a; Mai & Johansson, 2021), showing the potential of clipping operation in gradient rescaling for machine learning problems.

As far as we know, there is still no report on the application of gradient clipping in adversarial attacks. On the other hand, we empirically find that rescaling each past gradient in the momentum with its $L_1$-norm plays a key role in preserving a high success rate of attacks for MI-FGSM (Dong et al., 2018). Motivated by these facts, we will employ the gradient clipping technique to take the place of sign operation in MI-FGSM. Unfortunately, the normalized-gradient causes some problems in convergence analysis, which can also explain why the convergence for MI-FGSM and NI-FGSM has not been established so far. To overcome this difficulty, we present a new radius-varying clipping rule to guarantee its convergence. The use of the normalized-gradient with a radius-varying clipping rule is an obvious difference from the existing clipping-like algorithms. The contributions in this paper can be summarized as follows,

- We present a new clipped momentum method (called clipped NGM) for adversarial attacks, in which the normalized-gradient momentum (NGM) is clipped as its update direction with a radius-varying clipping rule.
- We prove that our clipped NGM attains its optimal averaging convergence for general constrained convex problems, indicating that the clipping strategy theoretically guarantees the stability of the whole optimization process and then fill the theory-practice gap for momentum-based adversarial attacks.
- The experiments verify that our clipped NGM can remarkably improve the performance of the state-of-the-art gradient-based algorithms in adversarial attacks, showing that the clipping technique can empirically serve as an alternative to the sign operation.

## 2 RELATED WORK

In this section, we provide a brief overview of several typical gradient-based methods for optimization and adversarial attacks.

### 2.1 MOMENTUM AND ITS CLIPPING

Consider
$$\min f(\boldsymbol{w}), \ s.t. \ \boldsymbol{w} \in \mathbf{Q}, \tag{1}$$
where $\mathbf{Q} \subseteq \mathbb{R}^N$ is a closed convex set and $f$ is a convex function on $\mathbf{Q}$. Assume that $\boldsymbol{w}^*$ is an optimal solution and $\nabla f(\boldsymbol{w})$ is a subgradient of $f$ at $\boldsymbol{w}$.

One of the most simple methods for solving the problem (1) is the projected subgradient (PSG) algorithm (Dimitri P. et al., 2003). Its key iteration is
$$\boldsymbol{w}_{t+1} = P_{\mathbf{Q}}[\boldsymbol{w}_t - \alpha_t \nabla f(\boldsymbol{w}_t)], \tag{2}$$
where $P_{\mathbf{Q}}$ is the projection operator on $\mathbf{Q}$ (Dimitri P. et al., 2003). By selecting suitable $\alpha_t > 0$, the online PSG achieves a data-dependant $O(\sqrt{t})$ regret bound (Zinkevich, 2003). With a standard

online-to-batch conversion, PSG attains an optimal averaging convergence rate of $O(\frac{1}{\sqrt{t}})$ for general convex functions, i.e., $f(\frac{1}{t}\sum_{k=1}^{t} \boldsymbol{w}_k) - f(\boldsymbol{w}^*) \leq O(\frac{1}{\sqrt{t}})$.

The regular Polyak's HB (Ghadimi et al., 2015) (Tao et al., 2021) for solving problem (1) is

$$\boldsymbol{w}_{t+1} = P_{\mathbf{Q}}[\boldsymbol{w}_t - \alpha_t \nabla f(\boldsymbol{w}_t) + \beta_t(\boldsymbol{w}_t - \boldsymbol{w}_{t-1})]. \tag{3}$$

With suitable $\alpha_t$ and $\beta_t$, it can achieve an optimal individual convergence rate, i.e., $f(\boldsymbol{w}_t) - f(\boldsymbol{w}^*) \leq O(\frac{1}{\sqrt{t}})$ (Tao et al., 2021). When $\mathbf{Q} = \mathbb{R}^N$, HB (3) can be rewritten as a two-steps algorithm

$$\begin{cases} \boldsymbol{m}_t = \beta_t \, \boldsymbol{m}_{t-1} + \alpha_t \, \nabla f(\boldsymbol{w}_t) \\ \boldsymbol{w}_{t+1} = \boldsymbol{w}_t - \boldsymbol{m}_t \end{cases}. \tag{4}$$

In deep learning community, HB in the form of (4) is usually referred to as momentum method (Ruder, 2016). In real applications, the momentum with an exponential moving average (EMA) (Tieleman & Hinton, 2012) (Kingma & Ba, 2015) is popularly used instead of $\boldsymbol{m}_t = \beta_t \, \boldsymbol{m}_{t-1} + \alpha_t \, \nabla f(\boldsymbol{w}_t)$ in (4), i.e.,

$$\boldsymbol{m}_t = \beta_t \, \boldsymbol{m}_{t-1} + (1 - \beta_t) \, \nabla f(\boldsymbol{w}_t). \tag{5}$$

Based on the EMA momentum (5), the sign-momentum (Bernstein et al., 2018) can be described as

$$\begin{cases} \boldsymbol{m}_t = \beta_t \, \boldsymbol{m}_{t-1} + (1 - \beta_t) \, \nabla f(\boldsymbol{w}_t) \\ \boldsymbol{w}_{t+1} = P_{\mathbf{Q}}[\boldsymbol{w}_t - \alpha_t \, \text{sign}(\boldsymbol{m}_t)] \end{cases}. \tag{6}$$

Specifically when $\beta_t \equiv 0$, (6) becomes the signGD. Note that the $i$-th component of $\text{sign}(\nabla f(\boldsymbol{w}))$ is $\frac{\nabla_i f(\boldsymbol{w})}{|\nabla_i f(\boldsymbol{w})|}$. Generally, the direction of $\nabla f(\boldsymbol{w})$ is different from that of $\text{sign}(\nabla f(\boldsymbol{w}))$. When $f$ is smooth without constraints, its convergence for nonconvex problems has been established under some standard assumptions (Bernstein et al., 2018).

When momentum clipping applied to (5) (Zhang et al., 2020a) (Mai & Johansson, 2021), it takes the form

$$\begin{cases} \boldsymbol{m}_t = \beta_t \, \boldsymbol{m}_{t-1} + (1 - \beta_t) \, \nabla f(\boldsymbol{w}_t) \\ \boldsymbol{w}_{t+1} = P_{\mathbf{Q}}[\boldsymbol{w}_t - \alpha_t \, \text{clip}_\gamma(\boldsymbol{m}_t)\boldsymbol{m}_t] \end{cases}, \tag{7}$$

where $\text{clip}_\gamma : \mathbb{R}^N \to \mathbb{R}$ and

$$\text{clip}_\gamma(\boldsymbol{w}) = \min\{1, \frac{\gamma}{\|\boldsymbol{w}\|}\}. \tag{8}$$

Here $\gamma$ is the clipping-radius and $\|\cdot\|$ represents the $L_2$-norm. It is easy to find that $\text{clip}_\gamma(\boldsymbol{w})$ is in fact the projection of $\boldsymbol{w}$ on the ball $\{\boldsymbol{w} : \|\boldsymbol{w}\| \leq \gamma\}$. Obviously, $\text{clip}_\gamma(\nabla f(\boldsymbol{w}))$ will not change the direction of $\nabla f(\boldsymbol{w})$ and only shrinks its norm when $\|\boldsymbol{w}\| \geq \gamma$. For optimizing non-convex but $(L_0, L_1)$-smooth functions, a tight convergence has been given in (Zhang et al., 2020a) showing that clipping algorithm has the same order of complexity as SGD (Ghadimi & Lan, 2013) but with a smaller factor. When $f$ is convex, assume that $f$ is quadratic growth, i.e., $f(\boldsymbol{w}) - f(\boldsymbol{w}^*) \geq \mu \min_{\boldsymbol{w}^* \in \mathbf{W}^*} \|\boldsymbol{w} - \boldsymbol{w}^*\|$, it holds that $\min_{\boldsymbol{w}^* \in \mathbf{W}^*} \|\boldsymbol{w} - \boldsymbol{w}^*\| \to 0$ under some regular conditions (Mai & Johansson, 2021), where $\mu > 0$ and $\mathbf{W}^*$ is the solution set of (1).

## 2.2 GRADIENT-BASED ATTACK METHODS

For a given classifier $f_{\boldsymbol{w}}$ with a predefined $\boldsymbol{w}$, generating an adversarial example $\boldsymbol{x}^{adv}$ from a real example $\boldsymbol{x}$ is usually formulated as the following constrained optimization problem (Madry et al., 2018; Goodfellow et al., 2015),

$$\max J(\boldsymbol{x}^{adv}, y), \; s.t. \; \|\boldsymbol{x}^{adv} - \boldsymbol{x}\|_\infty \leq \epsilon, \tag{9}$$

where $\mathcal{B}_\epsilon(\boldsymbol{x}) = \{\boldsymbol{x}^{adv} : \|\boldsymbol{x}^{adv} - \boldsymbol{x}\|_\infty \leq \epsilon\}$ and $J(\boldsymbol{x}, y)$ is a differentiable loss function $w.r.t.$ $\boldsymbol{x}$.

FGSM (Goodfellow et al., 2015) can be regarded as the first gradient-based attack method. Its update is

$$\boldsymbol{x}^{adv} = \boldsymbol{x} + \epsilon \, \text{sign}(\nabla_{\boldsymbol{x}} J(\boldsymbol{x}, y)). \tag{10}$$

Obviously, FGSM (10) is a one-step signGD with step-size $\epsilon$. By choosing the step-size $\epsilon$, we can easily get $\|\boldsymbol{x}^{adv} - \boldsymbol{x}\|_\infty \leq \epsilon$.

I-FGSM (Kurakin et al., 2017) is a specific signGD with step-size $\alpha$. It can be described as

$$\boldsymbol{x}_{t+1}^{adv} = \boldsymbol{x}_t^{adv} + \alpha \operatorname{sign}(\nabla_{\boldsymbol{x}_t^{adv}} J(\boldsymbol{x}_t^{adv}, y)), \tag{11}$$

where $\boldsymbol{x}_0^{adv} = \boldsymbol{x}$. The step-size is usually restricted to $\alpha = \epsilon/T$ to guarantee $\|\boldsymbol{x}_t^{adv} - \boldsymbol{x}\|_\infty \leq \epsilon$, where $T$ is the total number of iterations.

PGD (Carlini & Wagner, 2017) (Madry et al., 2018) is a sign-gradient PSG but starting from a random perturbation around the natural example, i.e.,

$$\boldsymbol{w}_{t+1} = P_{\mathcal{B}_\epsilon}[\boldsymbol{w}_t + \alpha_t \operatorname{sign}(\nabla_{\boldsymbol{x}_t^{adv}} J(\boldsymbol{x}_t^{adv}, y))], \tag{12}$$

Since the projection operator is used in PGD, it is not necessary to restrict the step-sized like I-FGSM. Specifically, for an image $\boldsymbol{x}^{adv} = (x_1^{adv}, x_2^{adv}, x_3^{adv})$ which is typically 3-D tensor, its projection is (Kurakin et al., 2017)

$$\begin{aligned}
&P_{\boldsymbol{x}}^\epsilon(\boldsymbol{x}^{adv}(x_1^{adv}, x_2^{adv}, x_3^{adv})) \\
&= \min\{255, \boldsymbol{x}(x_1, x_2, x_3) + \epsilon, max\{0, \boldsymbol{x}(x_1, x_2, x_3) - \epsilon, \boldsymbol{x}^{adv}(x_1^{adv}, x_2^{adv}, x_3^{adv})\}\}.
\end{aligned} \tag{13}$$

MI-FGSM (Dong et al., 2018) extends I-FGSM to momentum cases,

$$\begin{cases}
\boldsymbol{m}_t = \mu \, \boldsymbol{m}_{t-1} + \dfrac{\nabla_{\boldsymbol{x}_t^{adv}} J(\boldsymbol{x}_t^{adv}, y)}{\|\nabla_{\boldsymbol{x}_t^{adv}} J(\boldsymbol{x}_t^{adv}, y)\|_1} \\
\boldsymbol{x}_{t+1}^{adv} = \boldsymbol{x}_t^{adv} + \alpha \operatorname{sign}(\boldsymbol{m}_t)
\end{cases}, \tag{14}$$

where $\mu$ is the decay factor with $\boldsymbol{m}_0 = 0$ and $\alpha = \epsilon/T$. In contrast to the regular HB (4), each past gradient in $\boldsymbol{m}_t$ is normalized by its $L_1$-norm. To make a difference, we call this modification a normalized-gradient momentum (NGM). With such accumulation, its update direction $\operatorname{sign}(\boldsymbol{m}_t)$ is stabilized and then the transferability of adversarial examples is boosted (Dong et al., 2018).

MI-FGSM has achieved higher success rates in both white-box and black-box attacks than other algorithms, such as I-FGSM and PGD. This may be attributed to incorporating momentum with normalized gradients, which helps to improve stability. Nevertheless, MI-FGSM still belongs to the sign regime, which means it inevitably loses information about both the magnitude and direction of momentum. In order to overcome this limitation, we incorporate the clipping technique into MI-FGSM. In other words, our main idea in this work is to demonstrate, both theoretically and empirically, that the addition of the clipping step can take the place of sign operation especially in adversarial attacks.

## 3 THE PROPOSED CLIPPED NGM AND ITS OPTIMAL CONVERGENCE

Generally, the motivation behind the proposed clipped NGM is mainly derived from two facts. One is the practical success of NGM in MI-FGSM. The other is the advantage of clipping technique over sign operation.

For solving optimization problem (1), if the clipping technique is used to take the place of sign operation in MI-FGSM (14), we can get

$$\begin{cases}
\boldsymbol{m}_t = \mu \, \boldsymbol{m}_{t-1} + \dfrac{\nabla f(\boldsymbol{w}_t)}{\|\nabla f(\boldsymbol{w}_t)\|_1} \\
\boldsymbol{w}_{t+1} = P_{\mathbf{Q}}[\boldsymbol{w}_t - \alpha_t \operatorname{clip}_\gamma(\boldsymbol{m}_t)\boldsymbol{m}_t]
\end{cases}, \tag{15}$$

To emphasize the more recent gradients, we use the momentum in the form of EMA. (15) becomes

$$\begin{cases}
\boldsymbol{m}_t = \beta_t \, \boldsymbol{m}_{t-1} + (1 - \beta_t) \dfrac{\nabla f(\boldsymbol{w}_t)}{\|\nabla f(\boldsymbol{w}_t)\|_1} \\
\boldsymbol{w}_{t+1} = P_{\mathbf{Q}}[\boldsymbol{w}_t - \alpha_t \min\{1, \frac{\gamma}{\|\boldsymbol{m}_t)\|}\}\boldsymbol{m}_t]
\end{cases}. \tag{16}$$

Note that the regular Adam in (Kingma & Ba, 2015) can be described as

$$\begin{cases}
\boldsymbol{m}_t = \beta_{1t} \, \boldsymbol{m}_{t-1} + (1 - \beta_{1t})\nabla f(\boldsymbol{w}_t) \\
V_t = \beta_{2t} V_{t-1} + (1 - \beta_{2t})\operatorname{diag}(\nabla f(\boldsymbol{w}_t)\nabla f(\boldsymbol{w}_t)^\top) \\
\boldsymbol{w}_{t+1} = P_{\mathbf{Q}}[\boldsymbol{w}_t - \alpha_t V_t^{-\frac{1}{2}} \boldsymbol{m}_t]
\end{cases}. \tag{17}$$

In appearance, (16) is similar to Adam. In fact, the clipped EMA momentum can be regarded as a specific Adam but using the adaptive stepsize $V_t^{-\frac{1}{2}} = \min\{1, \frac{\gamma}{\|\boldsymbol{m}_t\|}\} I_N$. Unfortunately, it has been pointed out that Adam suffers from the non-convergence issue (Reddi et al., 2018). This fact indicates that the non-convergence (16) can be caused by the normalized-gradient.

Note that for a general convex $f$, we usually choose the monotonically decreasing step-size $\alpha_t = \frac{\alpha}{\sqrt{t}}$ to get optimal convergence of PSG (2) (Zinkevich, 2003). To guarantee the convergence of momentum methods with EMA, we can select $\beta_t = \beta_1 \lambda^{t-1}$ with $0 < \lambda < 1$ (Reddi et al., 2018). Based upon these facts, we propose a new clipped momentum method (called clipped NGM), the detailed steps of which are given in Algorithm 1.

---

**Algorithm 1** Clipped NGM

---

**Input:** The step-size parameter $\alpha > 0$; momentum parameters $o \leq \beta_1 < 1$ and $0 < \lambda < 1$; clipping parameter $0 < \gamma \leq 1$ and total number of iteration $T$.
1: Initialize $\boldsymbol{m}_0 = \boldsymbol{0}$ and $d_0 = 1$.
2: **repeat**
3:      $\boldsymbol{m}_t = \beta_t \boldsymbol{m}_{t-1} + (1 - \beta_t) \frac{\nabla f(\boldsymbol{w}_t)}{\|\nabla f(\boldsymbol{w}_t)\|_1}$
4:      $\alpha_t = \frac{\alpha}{\sqrt{t}}$
5:      $\beta_t = \beta_1 \lambda^{t-1}$
6:      $d_t = \min\{\frac{\gamma}{\|\boldsymbol{m}_t\|}, d_{t-1}\}$
7:      $\boldsymbol{w}_{t+1} = P_{\mathbf{Q}}[\boldsymbol{w}_t - \alpha_t d_t \boldsymbol{m}_t]$
8: **until** $t = T$
**Output:** $\boldsymbol{w}_{T+1}$.

---

Obviously, a main difference from MI-FGSM is that we employ clipping technique instead of the sign operation in Algorithm 1. In contrast to the regular clipped momentum (7), we use NGM with a new clipping strategy. To clearly understand our new clipping rule in Algorithm 1, we give

**Lemma 3.1.** *Let $\{d_t\}_{t=1}^T$ be the sequence generated by Algorithm 1. Suppose $0 < \gamma \leq 1$. Then we have*

$$d_t \geq \gamma, \ \forall t \geq 1.$$

According to Algorithm 1, Lemma 3.1 tells us

$$\gamma \leq d_t \leq d_{t-1} \leq 1, \ \forall t \geq 1,$$

which implies $d_t = \min\{\frac{\gamma}{\|\boldsymbol{m}_t\|}, d_{t-1}\} = \min\{1, \frac{\gamma}{\|\boldsymbol{m}_t\|}, d_{t-1}\}$, i.e., compared with the regular clipping operator (8), there is an extra requirement $d_t \leq d_{t-1}$ in Algorithm 1. Further,

$$d_t = \begin{cases} d_{t-1}, & \text{if } \|\boldsymbol{m}_t\| \leq \dfrac{\gamma}{d_{t-1}} \\[2mm] \dfrac{\gamma}{\|\boldsymbol{m}_t\|}, & \text{if } \|\boldsymbol{m}_t\| \geq \dfrac{\gamma}{d_{t-1}} \end{cases}, \tag{18}$$

which means that the clipping radius is $\frac{\gamma}{d_{t-1}}$, i.e., the clipping radius is increased since $\frac{\gamma}{d_{t-1}} \geq \gamma$. On the other hand, even if $\|\boldsymbol{m}_t\| \leq \frac{\gamma}{d_{t-1}}$, we still shrink $\|\boldsymbol{m}_t\|$ due to $\gamma \leq d_{t-1} \leq 1$. In general, the proposed clipping rule in Algorithm 1 can shrink $\|\boldsymbol{m}_t\|$ when its magnitude is big while keeping the direction of always unchanged.

To discuss the convergence of Algorithm 1, throughout this paper, we need several assumptions.

**Assumption 3.2.** *Assume that there exists a constant $G > 0$ such that*

$$\|\nabla f(\boldsymbol{w})\| \leq \|\nabla f(\boldsymbol{w})\|_1 \leq G, \ \forall \boldsymbol{w} \in \mathbf{Q}.$$

**Assumption 3.3.** *Assume that there exists a constant $D > 0$ such that*

$$\|\boldsymbol{w}_1 - \boldsymbol{w}_2\| \leq D, \forall \boldsymbol{w}_1, \boldsymbol{w}_2 \in \mathbf{Q}.$$

**Theorem 3.4.** *Let Assumption 3.2 and 3.3 hold and let $\{\boldsymbol{w}_t\}_{t=1}^T$ be generated by Algorithm 1. Suppose $0 < \beta_1 < 1$, $0 < \lambda < 1$ and $0 < \gamma \le 1$. Then we have*

$$f(\bar{\boldsymbol{w}}_T) - f(\boldsymbol{w}^*) \le \frac{GD^2}{2\alpha\gamma(1-\beta_1)\sqrt{T}} + \frac{GD^2}{2\alpha\gamma(1-\beta_1)(1-\lambda)^2 T} + \frac{2\alpha\gamma G}{(1-\beta_1)\sqrt{T}},$$

where $\bar{\boldsymbol{w}}_T = \frac{1}{T}\sum_{t=1}^T \boldsymbol{w}_t$.

The detailed proof Lemma 3.1 and Theorem 3.4 will be given in Appendix A. Obviously, Theorem 3.4 indicates that the clipped NGM achieves optimal averaging convergence for general convex problems, which avoids the non-convergence issue caused by the sign operation.

In particular, when $\beta_1 = 0$, the clipped NGM becomes

$$\boldsymbol{w}_{t+1} = P_{\mathbf{Q}}[\boldsymbol{w}_t - \alpha_t \min\{\frac{\gamma\|\nabla f(\boldsymbol{w}_t)\|_1}{\|\nabla f(\boldsymbol{w}_t)\|}, d_{t-1}\} \frac{\nabla f(\boldsymbol{w}_t)}{\|\nabla f(\boldsymbol{w}_t)\|_1}]. \tag{19}$$

Specifically when $\gamma = 1$, we have $\frac{\gamma}{\|\boldsymbol{m}_t\|} \ge 1$. According to Algorithm 1, it holds that $d_t \equiv 1$. The key iteration of clipped NGM becomes

$$\boldsymbol{w}_{t+1} = P_{\mathbf{Q}}[\boldsymbol{w}_t - \frac{\alpha}{\sqrt{t}}\boldsymbol{m}_t], \tag{20}$$

which means we do not conduct the clipping operation. For convenience and comparison in the experiments, we refer to (19) and (20) as clipped NG and NGM respectively.

## 4 EXPERIMENTS

In this section, we provide empirical evidence to illustrate the advantages of our clipped NGM over sign-like methods. Typically, by solving the optimization problem (9), we compare clipped NGM with the state-of-the-art I-FGSM, PGD and MI-FGSM, in which the projection operator (13) is used. Besides, NGM without clipping (NGM (20)) and the clipped method without momentum (Clipped NG (19)), Adam (Kingma & Ba, 2015), clipped-M (Mai & Johansson, 2021) are also considered as the baselines. Since our algorithm is directly established upon MI-FGSM, we first focus on comparing with MI-FGSM by conducting the same experiments. Comparison with other recent algorithms such as NI-FGSM, VMI-FGSM (Wang & He, 2021) on different network architectures can be seen in the Appendix.

### 4.1 DATASETS, MODELS AND PARAMETERS

We use the same dataset as that in (Dong et al., 2018; Lin et al., 2020), i.e., 1000 images covering 1000 categories are randomly selected from ILSVRC 2012 validation set (Russakovsky et al., 2015). Like (Dong et al., 2018; Lin et al., 2020), the attacked models cover both normally and adversarially trained models. For normally trained models, we consider Inception-v3 (Inc-v3) (Szegedy et al., 2016), Inception-v4 (Inc-v4), Inception-Resnet-v2 (Res-v2) (Szegedy et al., 2017) and ResNet-v2-101 (Res-101) (Ilyas et al., 2018). For adversarially trained models, we select Inc-v3ens3(v3ens3), Inc-v3ens4(v3ens4), IncRes-v2ens(v2ens) and Inc-v3Adv(v3Adv)(Tramèr et al., 2018).

Among all the experiments, the parameters of the optimization problem (9) are fixed, i.e., the maximum perturbation is $\epsilon = 16$ and the total number of iteration is $T = 10$ (Dong et al., 2018). For I-FGSM and MI-FGSM, as stated in Section 2.2, the constant step-size is set to $\alpha = \epsilon/T$ to make the generated adversarial examples satisfy the $l_\infty$ ball constraints, and the decay factor in MI-FGSM $\mu = 1$ (Dong et al., 2018). As usual, we set $\beta_1 = 0.9$ and $\lambda = 0.99$ in NGM and clipped NGM.

### 4.2 SELECTING $\alpha$ AND $\gamma$

In contrast to setting the constant learning rate and constant clipping radius in the existing algorithms, we essentially adopt the time-varying learning rate $\alpha_t$ and time-varying clipping radius $d_t$ for clipped NGM in all our experiments, which completely coincides with our theoretical analysis

in 3.4. Note that only two important parameters left to be adjusted, i.e., the step-size parameter $\alpha$ and clipping parameter $\gamma$.

To select suitable $\alpha$ and $\gamma$, we use clipped NGM to attack Inc-v3 by grid search, in which $\alpha$ ranges from 2 to 30 and $\gamma$ ranges from 0.1 to 1.0. We show the success rates of the generated adversarial examples against Inc-v3 (white-box) and seven black-box attacks in Fig.1. Considering all the black-box attacks, we set $\alpha = 18$ and $\gamma = 0.8$ for our clipped NGM throughout all the experiments.

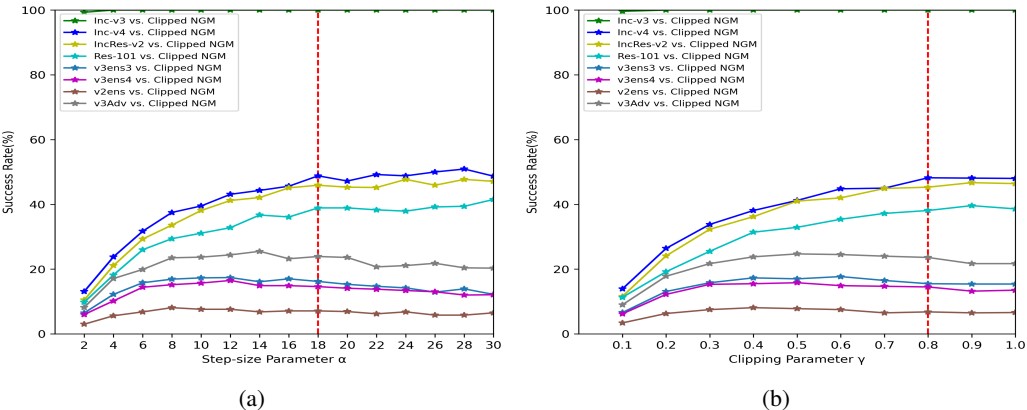

Figure 1: Attack success rates (%) of the adversarial examples generated for Inc-v3 (white-box) against Inc-v4, IncRes-v2, Res-101, v3ens3, v3ens4, v2ens and v3Adv (black-box). (a) $\alpha = 0.8$. (b) $\gamma = 18$.

## 4.3 ATTACKING A SINGLE MODEL

The success rates of attacks against normally and adversarially trained models are reported in Table 1. From this table, several interesting phenomena can be observed. First of all, it is not surprising at all that the performance of Adam and Clipped-M is even worse than MI-FGSM. One cause for such failures is that the normalized gradient is not used in these algorithms. Secondly, NGM obtains the best success rate on each white-box attack. However, it is consistently inferior to MI-FGSM on all the black-box attacks. Finally, clipped NG gets a better success rate than that of MI-FGSM on each adversarially trained model. Unfortunately, it is consistently inferior to MI-FGSM on all the normally trained models. Fortunately, we observe that our clipped NGM achieves the best success rates for almost all the attacks among all the algorithms, demonstrating the advantages over sign-like algorithms and their baselines.

## 4.4 STABILITY OF CLIPPED NGM

It has been indicated that MI-FGSM can stabilize update directions due to integrating the momentum term into the iterative process for attacks (Dong et al., 2018). To make a comparison between MI-FGSM and clipped NGM, we investigate the changing behaviour of success rates with respect to the number of iterations. Typically, we only consider generating adversarial examples on Inc-v3.

As can be seen in Fig.(2), when the number of iterations increases, both MI-FGSM and clipped NGM can obtain a near 100% success rate on white-box attacks. Note in Section 4.5, we only report the success rates of each concerned algorithm at $T = 10$ like that in (Dong et al., 2018). From Fig.(2), we indeed find that the overall best black-box success rate of MI-FGSM is obtained when $T$ approaches 10. However, this success rate will decrease when $t$ becomes larger. Fortunately, our clipped NGM can maintain the success rate in a relatively high level even when $T > 10$, exhibiting its better stability.

## 4.5 ATTACKING AN ENSEMBLE OF MODELS

It has been shown that attacking multiple models at the same time can improve the transferability of generated adversarial examples (Liu et al., 2017). To further compare with I-FGSM, PGD and MI-FGSM, we apply clipped NGM to attack an ensemble of models. As pointed out in (Dong et al., 2018), the ensemble in logits outperforms the ensemble in predictions and the ensemble in

Table 1: Attack success rates (%) of adversarial attacks against baseline methods. The adversarial examples are crafted on Inc-v3, Inc-v4, IncRes-v2, and Res-101 respectively using I-FGSM, PGD, MI-FGSM, NGM, clipped NG and clipped NGM. * indicates the white-box attacks.

| Model | Attack | Inc-v3 | Inc-v4 | IncRes-v2 | Res-101 | v3ens3 | v3ens4 | v2ens | v3Adv |
|---|---|---|---|---|---|---|---|---|---|
| Inc-v3 | I-FGSM | 100.0* | 22.3 | 18.0 | 15.0 | 5.1 | 5.1 | 2.6 | 6.5 |
| | PGD | 100.0* | 31.3 | 26.8 | 23.6 | 6.6 | 8.0 | 4.2 | 9.1 |
| | Adam | 99.5* | 38.6 | 35.6 | 29.1 | 12.9 | 11.4 | 6.2 | 16.3 |
| | MI-FGSM | 100.0* | 44.3 | 42.1 | 36.1 | 14.3 | 13.7 | 6.2 | 18.9 |
| | NGM | 100.0 * | 38.8 | 36.0 | 31.5 | 8.1 | 7.7 | 3.2 | 20.6 |
| | Clipped-M | 96.5* | 42.9 | 40.6 | 34.1 | 10.9 | 10.1 | 5.6 | 17.7 |
| | Clipped NG | 100.0 * | 39.2 | 37.5 | 29.0 | 15.8 | 14.4 | 6.8 | 21.9 |
| | **Clipped NGM** | **100.0 *** | **48.8** | **45.9** | **38.9** | **16.2** | **14.6** | **7.1** | **23.9** |
| Inc-v4 | I-FGSM | 32.1 | 99.8* | 20.5 | 19.6 | 6.0 | 6.4 | 3.0 | 6.3 |
| | PGD | 38.4 | 100.0* | 28.5 | 26.3 | 7.5 | 7.4 | 5.0 | 9.0 |
| | Adam | 48.7 | 99.6* | 39.8 | 35.5 | 16.0 | 13.7 | 7.5 | 15.1 |
| | MI-FGSM | 56.8 | 99.7* | 46.4 | **42.5** | 16.6 | 14.7 | 8.0 | 18.4 |
| | NGM | 53.5 | 100.0* | 40.2 | 36.5 | 8.6 | 8.5 | 4.0 | 19.2 |
| | Clipped-M | 51.3 | 90.2 * | 43.2 | 40.4 | 15.1 | 13.6 | 6.8 | 20.0 |
| | Clipped NG | 48.9 | 100.0* | 38.0 | 33.7 | 17.4 | 14.9 | **9.7** | 19.5 |
| | **Clipped NGM** | **57.3** | 99.9* | **47.8** | 42.1 | **19.2** | **15.0** | 7.6 | **21.5** |
| IncRes-v2 | I-FGSM | 33.7 | 25.6 | 98.3 * | 20.2 | 7.8 | 6.3 | 4.4 | 7.9 |
| | PGD | 41.1 | 32.6 | 100.0* | 27.0 | 8.6 | 8.1 | 6.0 | 10.6 |
| | Adam | 54.8 | 46.1 | 98.2* | 37.7 | 21.2 | 16.0 | 11.8 | 20.1 |
| | MI-FGSM | 60.0 | 50.6 | 98.0 * | **44.1** | 21.6 | 16.4 | 11.3 | 23.0 |
| | NGM | 56.9 | 42.1 | 99.2 * | 35.0 | 9.6 | 8.6 | 4.6 | 22.4 |
| | Clipped-M | 47.4 | 41.9 | 79.2* | 35.1 | 22.9 | 16.4 | 13.3 | 23.4 |
| | Clipped NG | 51.1 | 44.0 | 98.7* | 34.9 | 22.2 | 16.4 | 14.4 | 23.2 |
| | **Clipped NGM** | **60.0** | **52.9** | 99.0* | 44.0 | **24.3** | **18.8** | **13.7** | **29.0** |
| Res-101 | I-FGSM | 30.9 | 25.1 | 23.2 | 99.3 * | 8.0 | 7.6 | 4.6 | 9.0 |
| | PGD | 43.9 | 35.1 | 33.7 | 99.3* | 10.8 | 10.1 | 6.7 | 11.0 |
| | Adam | 48.6 | 43.7 | 41.4 | 97.3* | 23.2 | 20.4 | 12.4 | 21.2 |
| | MI-FGSM | 56.6 | 51.6 | 48.7 | 99.3 * | 24.1 | 22.0 | 12.1 | 24.9 |
| | NGM | 55.9 | 47.9 | 46.7 | 99.5 * | 11.5 | 9.9 | 5.7 | 23.1 |
| | Clipped-M | 50.0 | 45.1 | 45.1 | 87.5 | 21.4 | 17.5 | 10.9 | 24.9 |
| | Clipped NG | 48.3 | 42.7 | 40.8 | 99.3* | 24.9 | 22.3 | 16.1 | 26.9 |
| | **Clipped NGM** | **59.6** | **53.5** | **51.4** | 99.3 * | **29.3** | **24.4** | **16.4** | **34.4** |

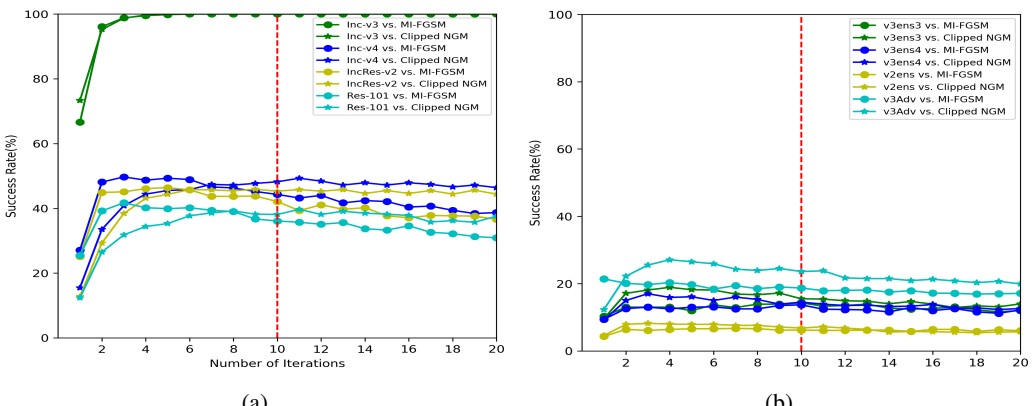

(a)                                                      (b)

Figure 2: Attack success rates (%) of the adversarial examples generated for Inc-v3 (white-box) against Inc-v4, IncRes-v2, Res-101, v3ens3, v3ens4, v2ens and v3Adv (black-box).

loss consistently among all the attack methods and different models in the ensemble for both the white-box and black-box attacks. Therefore, we only focus on attacking an ensemble of normally trained models in logits (including Inc-v3, Inc-v4, IncRes-v2 and Res-101) with equal weights. We report the success rates of attack against adversarially trained models in Table 2.

By comparing the experimental results in Table 1 and 2, it is easy to find that our clipped NGM under the multi-model setting can similarly improve the transferability. Fortunately, our clipped NGM remarkably outperforms I-FGSM, PGD and MI-FGSM when attacking an ensemble of models.

Table 2: Success rates (%) of adversarial attacks under multi-model setting. * indicates the white-box attacks.

| Attack | Inc-v3* | Inc-v4* | IncRes-v2* | Res-101* | v3ens3 | v3ens4 | v2ens | v3Adv |
|--------|---------|---------|------------|----------|--------|--------|-------|-------|
| I-FGSM | 100.0 | 100.0 | 99.6 | 100.0 | 19.1 | 16.7 | 9.0 | 16.2 |
| PGD | 100.0 | 99.6 | 98.9 | 99.8 | 20.7 | 16.9 | 10.4 | 17.7 |
| MI-FGSM | 100.0 | 99.6 | 99.4 | 99.9 | 48.2 | 42.9 | 27.9 | 43.2 |
| **Clipped NGM** | 100.0 | 99.9 | 99.8 | 100.0 | **53.6** | **47.3** | **33.2** | **59.7** |

### 4.6 VISUALIZATION OF ADVERSARIAL EXAMPLES

We show 6 randomly selected benign images and their corresponding adversarial images, which are generated by I-FGSM, PGD, MI-FGSM and clipped NGM for Inc-v3. As can be seen in 3, whether the background is simple or complex, the difference between the noise images generated by MI-FGSM and our clipped NGM is almost human-imperceptible.

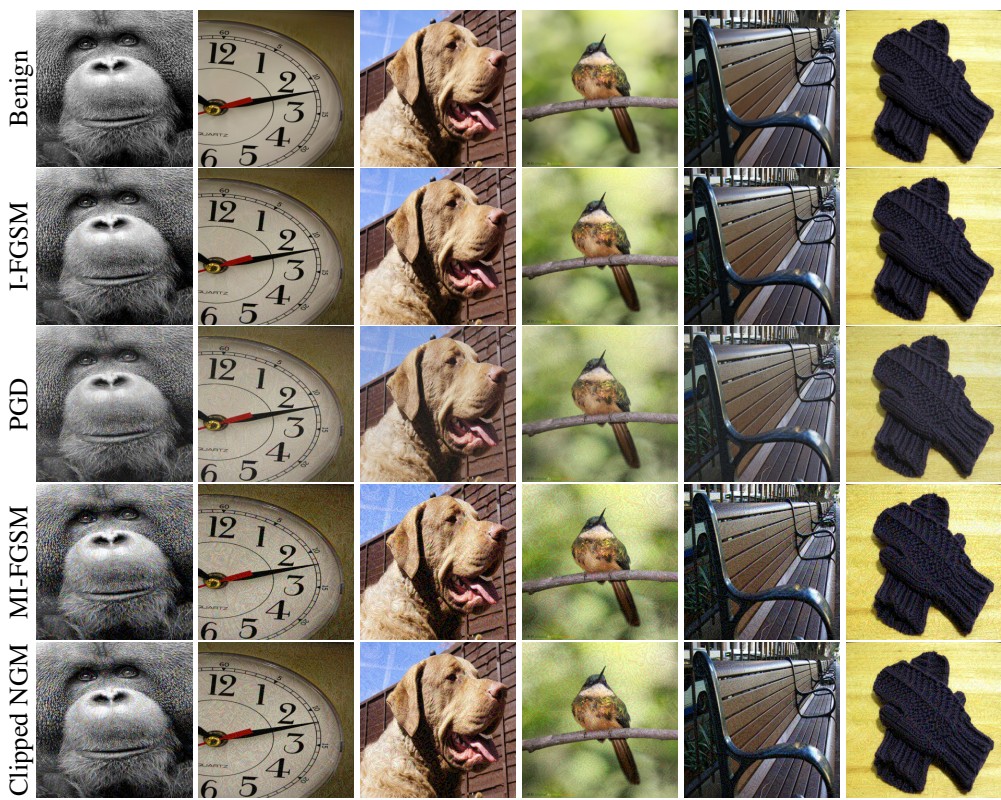

Figure 3: Visualization of randomly picked benign images and their corresponding adversarial images.

## 5 CONCLUSION AND FUTURE WORK

In this paper, we use clipping technique to overcome the limitations of sign operation in state-of-the-art gradient-based adversarial attack methods. To obtain better practical performance, the normalized-gradient momentum is employed. To guarantee convergence for general convex problems, the radius-varying clipping rule is proposed. Our clipped NGM fills the theory-practice gap for momentum-based attacks. From the viewpoint of both theoretical analysis and empirical study, our clipping technique can serve as an alternative to the sign operation in adversarial attacks.

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

# A  APPENDIX

## A.1  CONVERGENCE ANALYSIS OF CLIPPED NGM

**Proof of Lemma 3.1**

Note that $m_t$ is a convex combination of each $\frac{\nabla f(w_i)}{\|\nabla f(w_i)\|_1}$, and $\frac{\|\nabla f(w_i)\|}{\|\nabla f(w_i)\|_1} \le 1$ $(i = 1, 2, \cdots, t)$. We have

$$\|m_t\| \le 1.$$

Then $\frac{\gamma}{\|m_t\|}\| \ge \gamma$ and Lemma 3.1 can be obtained by simple induction.

**Proof of Theorem 3.4**

From the non-expansiveness of the projection operator, we know

$$
\begin{aligned}
\|w_{t+1} - w^*\|^2 &\le \|w_t - \alpha_t d_t m_t - w^*\|^2 \\
&= \|w_t - w^*\|^2 + \|\alpha_t d_t m_t\|^2 - 2\alpha_t d_t \langle m_t, w_t - w^* \rangle \\
&= \|w_t - w^*\|^2 + \|\alpha_t d_t m_t\|^2 - 2\alpha_t d_t \langle \beta_t m_{t-1} + (1 - \beta_t)\frac{\nabla f(w_t)}{\|\nabla f(w_t)\|_1}, w_t - w^* \rangle.
\end{aligned}
$$

Rearrange the inequality, we have

$$
\begin{aligned}
&\frac{2\alpha_t d_t (1 - \beta_t)}{\|\nabla f(w_t)\|_1} \langle \nabla f(w_t), w_t - w^* \rangle \\
&\le \|w_t - w^*\|^2 - \|w_{t+1} - w^*\|^2 + \|\alpha_t d_t m_t\|^2 - 2\alpha_t d_t \beta_t \langle m_{t-1}, w_t - w^* \rangle,
\end{aligned}
$$

i.e.,

$$
\begin{aligned}
\frac{1}{\|\nabla f(w_t)\|_1} \langle \nabla f(w_t), w_t - w^* \rangle &\le \frac{\|w_t - w^*\|^2 - \|w_{t+1} - w^*\|^2}{2\alpha_t d_t (1 - \beta_t)} + \frac{\alpha_t d_t \|m_t\|^2}{2(1 - \beta_t)} - \frac{\beta_t \langle m_{t-1}, w_t - w^* \rangle}{(1 - \beta_t)} \\
&\le \frac{\|w_t - w^*\|^2 - \|w_{t+1} - w^*\|^2}{2\alpha_t d_t (1 - \beta_t)} + \frac{\alpha_t d_t \|m_t\|^2}{2(1 - \beta_t)} + \frac{\alpha_t d_t \beta_t \|m_{t-1}\|^2}{2(1 - \beta_t)} \\
&\quad + \frac{\beta_t \|w_t - w^*\|^2}{2\alpha_t d_t (1 - \beta_t)}.
\end{aligned}
$$

Using the property of convex functions,

$$\langle \nabla f(w_t), w_t - w^* \rangle \ge f(w_t) - f(w^*).$$

Then

$$
\begin{aligned}
\frac{f(w_t) - f(w^*)}{G} &\le \frac{\|w_t - w^*\|^2 - \|w_{t+1} - w^*\|^2}{2\alpha_t d_t (1 - \beta_t)} + \frac{\alpha_t d_t \|m_t\|^2}{2(1 - \beta_t)} + \frac{\alpha_t d_t \beta_t \|m_{t-1}\|^2}{2(1 - \beta_t)} \\
&\quad + \frac{\beta_t \|w_t - w^*\|^2}{2\alpha_t d_t (1 - \beta_t)}.
\end{aligned}
$$

Summing this inequality from $t = 1$ to $T$, we obtain

$$
\begin{aligned}
\frac{1}{G}\sum_{t=1}^{T}[f(w_t) - f(w^*)] \le &\underbrace{\sum_{t=1}^{T}\frac{\|w_t - w^*\|^2 - \|w_{t+1} - w^*\|^2}{2\alpha_t d_t (1 - \beta_t)}}_{P_1} + \underbrace{\sum_{t=1}^{T}\frac{\beta_t \|w_t - w^*\|^2}{2\alpha_t d_t (1 - \beta_t)}}_{P_2} \\
&+ \underbrace{\sum_{t=1}^{T}\frac{\alpha_t d_t \|m_t\|^2}{2(1 - \beta_t)} + \sum_{t=1}^{T}\frac{\alpha_t d_t \beta_t \|m_{t-1}\|^2}{2(1 - \beta_t)}}_{P_3}.
\end{aligned}
$$

To bound $P_1$, we have

$$
\begin{aligned}
P_1 &= \sum_{t=1}^{T} \frac{\|\boldsymbol{w}_t - \boldsymbol{w}^*\|^2 - \|\boldsymbol{w}_{t+1} - \boldsymbol{w}^*\|^2}{2\alpha_t d_t (1 - \beta_t)} \\
&= \sum_{t=2}^{T} \left( \frac{1}{2\alpha_t d_t (1 - \beta_t)} - \frac{1}{2\alpha_{t-1} d_{t-1} (1 - \beta_{t-1})} \right) \|\boldsymbol{w}_t - \boldsymbol{w}^*\|^2 \\
&\quad + \frac{\|\boldsymbol{w}_1 - \boldsymbol{w}^*\|^2}{2\alpha_1 d_1 (1 - \beta_1)} - \frac{\|\boldsymbol{w}_{T+1} - \boldsymbol{w}^*\|^2}{2\alpha_T d_T (1 - \beta_T)} \\
&\leq \sum_{t=2}^{T} \left( \frac{1}{2\alpha_t d_t (1 - \beta_t)} - \frac{1}{2\alpha_{t-1} d_{t-1} (1 - \beta_{t-1})} \right) D^2 + \frac{D^2}{2\alpha_1 d_1 (1 - \beta_1)} \\
&= \frac{D^2}{2\alpha_T d_T (1 - \beta_T)} \leq \frac{D^2 \sqrt{T}}{2\alpha\gamma (1 - \beta_1)}.
\end{aligned}
\tag{21}
$$

To bound $P_2$, according to Lemma 3.1, we have

$$
\begin{aligned}
P_2 &= \sum_{t=1}^{T} \frac{\beta_t \|\boldsymbol{w}_t - \boldsymbol{w}^*\|^2}{2\alpha_t d_t (1 - \beta_t)} \leq \frac{1}{\gamma} \sum_{t=1}^{T} \frac{\beta_t \|\boldsymbol{w}_t - \boldsymbol{w}^*\|^2}{2\alpha_t (1 - \beta_t)} \leq \frac{D^2}{2\alpha\gamma} \sum_{t=1}^{T} \frac{\beta_t \sqrt{t}}{(1 - \beta_t)} \\
&\leq \frac{D^2}{2\alpha\gamma (1 - \beta_1)} \sum_{t=1}^{T} \lambda^{t-1} \sqrt{t} \leq \frac{D^2}{2\alpha\gamma (1 - \beta_1)} \sum_{t=1}^{T} \lambda^{t-1} t \leq \frac{D^2}{2\alpha\gamma (1 - \beta_1)(1 - \lambda)^2}.
\end{aligned}
\tag{22}
$$

To bound $P_3$, we have

$$
\begin{aligned}
P_3 &= \sum_{t=1}^{T} \frac{\alpha_t d_t \|\boldsymbol{m}_t\|^2}{2(1 - \beta_t)} + \sum_{t=1}^{T} \frac{\alpha_t d_t \beta_t \|\boldsymbol{m}_{t-1}\|^2}{2(1 - \beta_t)} \\
&\leq \sum_{t=1}^{T} \frac{\alpha_t d_t \|\boldsymbol{m}_t\|^2}{2(1 - \beta_t)} + \sum_{t=1}^{T} \frac{\alpha_{t-1} d_{t-1} \beta_t \|\boldsymbol{m}_{t-1}\|^2}{2(1 - \beta_t)} \\
&\leq \frac{1}{2(1 - \beta_1)} \sum_{t=1}^{T} \alpha_t d_t \|\boldsymbol{m}_t\|^2 + \frac{1}{2(1 - \beta_1)} \sum_{t=1}^{T} \alpha_{t-1} d_{t-1} \beta_t \|\boldsymbol{m}_{t-1}\|^2 \\
&\leq \frac{1}{(1 - \beta_1)} \sum_{t=1}^{T} \alpha_t d_t \|\boldsymbol{m}_t\|^2 \leq \frac{\alpha\gamma}{(1 - \beta_1)} \sum_{t=1}^{T} \frac{1}{\sqrt{t}} \leq \frac{2\alpha\gamma \sqrt{T}}{(1 - \beta_1)}.
\end{aligned}
\tag{23}
$$

Combining Equation (21), Equation (22) and Equation (23), we have

$$
\sum_{t=1}^{T} [f(\boldsymbol{w}_t) - f(\boldsymbol{w}^*)] \leq \frac{GD^2 \sqrt{T}}{2\alpha\gamma (1 - \beta_1)} + \frac{GD^2}{2\alpha\gamma (1 - \beta_1)(1 - \lambda)^2} + \frac{2\alpha\gamma G \sqrt{T}}{(1 - \beta_1)}.
$$

Thus

$$
\frac{1}{T} \sum_{t=1}^{T} [f(\boldsymbol{w}_t) - f(\boldsymbol{w}^*)] \leq \frac{GD^2}{2\alpha\gamma (1 - \beta_1)\sqrt{T}} + \frac{GD^2}{2\alpha\gamma (1 - \beta_1)(1 - \lambda)^2 T} + \frac{2\alpha\gamma G}{(1 - \beta_1)\sqrt{T}}.
$$

By convexity of $f(\boldsymbol{w})$, we obtain

$$
f(\bar{\boldsymbol{w}}_T) - f(\boldsymbol{w}^*) \leq \frac{GD^2}{2\alpha\gamma (1 - \beta_1)\sqrt{T}} + \frac{GD^2}{2\alpha\gamma (1 - \beta_1)(1 - \lambda)^2 T} + \frac{2\alpha\gamma G}{(1 - \beta_1)\sqrt{T}}.
\tag{24}
$$

Table 3: Attack success rates (%) of adversarial attacks against baseline methods. The adversarial examples are crafted on Inc-v3, Inc-v4, IncRes-v2, and Res-101 respectively using VMI-FGSM and Clipped VMI-FGSM. * indicates the white-box attacks.

| Model | Attack | Inc-v3 | Inc-v4 | IncRes-v2 | Res-101 | v3ens3 | v3ens4 | v2ens | v3Adv |
|-------|--------|--------|--------|-----------|---------|--------|--------|-------|-------|
| Inc-v3 | VMI-FGSM | 100.0* | 71.6 | 68.7 | 59.9 | 32.4 | 30.5 | 18.0 | 36.2 |
| | **Clipped VMI-FGSM** | 100.0 * | **75.2** | **74.9** | **65.2** | **34.5** | **32.1** | **18.3** | **46.7** |
| Inc-v4 | VMI-FGSM | 76.7 | 99.7* | 70.7 | 63.2 | 38.2 | 36.8 | 23.5 | 34.7 |
| | **Clipped VMI-FGSM** | **81.6** | 100.0 * | **75.3** | **67.1** | **43.1** | **39.0** | **25.2** | **45.6** |
| IncRes-v2 | VMI-FGSM | 77.8 | 72.2 | 97.9 * | 67.2 | 47.0 | 39.9 | 34.7 | 43.0 |
| | **Clipped VMI-FGSM** | **81.9** | **77.6** | 98.1 * | **70.8** | **53.5** | **45.2** | **39.0** | **56.6** |
| Res-101 | VMI-FGSM | 75.0 | 69.1 | 69.9 | 99.3 * | 45.5 | 42.1 | 30.5 | 42.9 |
| | **Clipped VMI-FGSM** | **79.1** | **74.0** | **72.9** | 99.5 * | **50.0** | **44.6** | **35.3** | **55.8** |

## A.2 COMPARISON WITH SOME RECENT ALGORITHMS

In Wang & He (2021), a variance tuning technique is proposed to enhance the class of iterative gradient based attack methods and improve their attack transferability. Specifically, at each iteration for the gradient calculation, instead of directly using the current gradient for the momentum accumulation, they use the gradient variance of the previous iteration to tune the current gradient so as to further stabilize the update direction and escape from poor local optima. When this technique is integrated into MI-FGSM, VMI-FGSM can be obtained Wang & He (2021).

As indicated in Wang & He (2021), the variance tuning technique is generally applicable to any gradient-based attack method. Similarly, by using the gradient variance of the previous iteration to tune the current gradient in clipped NGM, we can get clipped VMI-FGSM. The detailed steps of clipped VMI-FGSM are given in Algorithm 2, where $\boldsymbol{x}_t^{adv,i} = \boldsymbol{x}_t^{adv} + r_i, r_i \sim U[-(\beta \cdot \epsilon)^d, (\beta \cdot \epsilon)^d]$ with $\beta$ the upper bound of neighborhood, $d$ the dimensions. The success rates of attacks against normally and adversarially trained models are reported in Table 3. These experimental results can further verify that the sign-operation can be replaced by the clipping strategies.

---

**Algorithm 2** Clipped VMI-FGSM

---

**Input:** The step-size parameter $\alpha > 0$; momentum parameters $o \leq \beta_1 < 1$ and $0 < \lambda < 1$; clipping parameter $0 < \gamma \leq 1$; number of example for variance tuning $N$ and total number of iteration $T$.

1: Initialize $\boldsymbol{m}_0 = \boldsymbol{0}$ and $d_0 = 1$.
2: **repeat**
3: $\quad \boldsymbol{m}_t = \beta_t \boldsymbol{m}_{t-1} + (1 - \beta_t) \frac{\nabla_{\boldsymbol{x}_t^{adv}} J(\boldsymbol{x}_t^{adv}, y) + \boldsymbol{v}_t}{\|\nabla_{\boldsymbol{x}_t^{adv}} J(\boldsymbol{x}_t^{adv}, y) + \boldsymbol{v}_t\|_1}$
4: $\quad \boldsymbol{v}_{t+1} = \frac{1}{N} \sum_{i=1}^{N} \nabla_{\boldsymbol{x}_t^{adv,i}} J(\boldsymbol{x}_t^{adv,i}, y) - \nabla_{\boldsymbol{x}_t^{adv}} J(\boldsymbol{x}_t^{adv}, y)$
5: $\quad \alpha_t = \frac{\alpha}{\sqrt{t}}$
6: $\quad \beta_t = \beta_1 \lambda^{t-1}$
7: $\quad d_t = \min\{\frac{\gamma}{\|\boldsymbol{m}_t\|}, d_{t-1}\}$
8: $\quad \boldsymbol{x}_{t+1}^{adv} = P_{\mathbf{Q}}[\boldsymbol{x}_t^{adv} + \alpha_t d_t \boldsymbol{m}_t]$
9: **until** $t = T$
**Output:** $\boldsymbol{x}_{T+1}^{adv}$.

---

## A.3 COMPARISON ACROSS DIFFERENT ARCHITECTURES

Recently, RobustBench has become a widely recognized benchmark for the adversarial robustness of image classification networks. In its most commonly reported sub-task, RobustBench evaluates and ranks the adversarial robustness of trained neural networks under AutoAttack (Croce et al., 2020) (Croce & Hein, 2020). They open source the Model Zoo for convenient access to the 80+ most robust models from the literature which can be used for downstream tasks and facilitate the development of new standardized attacks. More recently, the adversarial transferability of adversarial examples

Table 4: Transferability with ViT-B/16 as the source model ($T = 10$).

| Attack | PGD | MI-FGSM | NI-FGSM | VMI-FGSM | Clipped NGM |
|---|---|---|---|---|---|
| VGG16 | 29.64 | 29.70 | 29.84 | 26.58 | **36.44** |
| Res50 | 20.64 | 20.42 | 20.84 | 18.56 | **25.02** |
| Dense121 | 20.36 | 21.16 | 20.92 | 18.90 | **18.59** |
| Inc-V3 | 15.06 | 15.94 | 16.08 | 14.14 | **18.59** |
| ViT-S/16 | 59.84 | 60.98 | 60.82 | 53.80 | **64.95** |
| ViT-B/16* | 99.98 | 100 | 100 | 99.20 | 99.95 |
| ViT-L/16 | 45.34 | 44.44 | 45.82 | 40.84 | **64.40** |
| Deit-T/16 | 43.30 | 42.40 | 43.04 | 38.90 | **44.42** |
| Deit-S/16 | 48.26 | 48.06 | 48.96 | 43.64 | **50.18** |
| Deit-B/16 | 50.42 | 49.38 | 50.98 | 44.90 | **51.38** |
| Swin-B/4/7 | 27.84 | 27.14 | 27.76 | 25.50 | **29.89** |
| SNN-VGG | 48.94 | 48.96 | 50.40 | 44.00 | **55.62** |
| SNN-ResNet | 72.56 | 72.18 | 74.60 | 67.66 | **74.46** |
| SNN-VGG | 53.48 | 54.12 | 55.50 | 48.28 | **61.58** |
| SNN-ResNet | 55.10 | 52.92 | 55.26 | 47.64 | **61.32** |
| DyNN-R50/96 | 13.38 | 12.54 | 13.02 | 12.08 | **15.68** |
| DyNN-R50/128 | 15.52 | 14.62 | 15.42 | 14.26 | **18.70** |
| DyNN-D121/96 | 13.9 | 13.56 | 13.98 | 13.08 | **17.08** |

Table 5: Transferability with VGG16 as the source model ($T = 10$).

| Attack | PGD | MI-FGSM | NI-FGSM | VMI-FGSM | Clipped NGM |
|---|---|---|---|---|---|
| VGG16* | 99.44 | 99.3 | 99.22 | 98.62 | 99.44 |
| Res50 | 19.82 | 40.96 | 41.76 | 40.02 | **51.94** |
| Dense121 | 16.70 | 38.96 | 39.72 | 37.52 | **46.64** |
| Inc-V3 | 7.52 | 23.00 | 22.84 | 21.62 | **28.84** |
| ViT-S/16 | 1.68 | 5.28 | 5.66 | 5.40 | **6.60** |
| ViT-B/16 | 1.10 | 3.46 | 3.34 | 2.28 | **3.84** |
| ViT-L/16 | 0.52 | 1.52 | 1.38 | 1.50 | **1.72** |
| Deit-T/16 | 4.78 | 12.88 | 12.30 | 12.04 | **14.84** |
| Deit-S/16 | 1.58 | 6.20 | 5.62 | 5.24 | **7.40** |
| Deit-B/16 | 1.18 | 3.16 | 3.38 | 3.22 | **3.94** |
| Swin-B/4/7 | 1.96 | 4.82 | 4.56 | 4.58 | **5.70** |
| SNN-VGG | 93.54 | 98.24 | 98.22 | 97.36 | **98.76** |
| SNN-ResNet | 47.90 | 73.36 | 72.84 | 72.16 | **76.76** |
| SNN-VGG/L | 92.62 | 98.02 | 98.08 | 97.30 | **98.78** |
| SNN-ResNet/L | 27.40 | 57.36 | 57.30 | 55.52 | **68.50** |
| DyNN-R50/96 | 5.40 | 10.84 | 11.36 | 11.54 | **13.46** |
| DyNN-R50/128 | 5.92 | 14.48 | 14.54 | 14.50 | **18.68** |
| DyNN-D121/96 | 5.64 | 11.26 | 11.36 | 11.50 | **13.32** |

generated with representative attacks is re-evaluated across different types of deep neural networks (Yu et al., 2023).

Since we only focus on evaluating the transferability of adversarial attack of gradient-based algorithms, we choose to compare our clipped NGM with all the mentioned gradient-based algorithms and models in (Yu et al., 2023) using their evaluation set, which includes PGD, MI-FGSM, NI-FGSM and VMI-FGSM. We use the same parameters ($\alpha$ and $\gamma$) decided by our previous CNN experiments. The experimental results are shown in Table 4, Table 5 and Table 6, in which the setting of other gradient-based algorithms is directly from (Yu et al., 2023).

From the above tables, we can observe that our clipped NGM almost consistently outperforms MI-FGSM when using ViT-B/16 as the source model. When VGG16 and IncV3 are employed as the source models, our clipped NGM outperforms MI-FGSM on most models. Note that the transferability within CNN has been sufficiently verified in our previous experiments. Since we use the same parameters in our algorithm, these experimental results further illustrate that the clipping technique has better transferability across deferent network architectures and then can serve as an alternative to the dominant sign operation in gradient-based adversarial attacks.

Table 6: Transferability with IncV3 as the source model ($T = 10$).

| Attack | PGD | MI-FGSM | NI-FGSM | VMI-FGSM | Clipped NGM-299 |
|---|---|---|---|---|---|
| VGG16 | 9.36 | 25.44 | 24.92 | 16.94 | **33.44** |
| Res50 | 5.96 | 17.28 | 16.80 | 12.40 | **21.08** |
| Dense121 | 5.64 | 16.42 | 16.40 | 12.46 | **18.54** |
| Inc-V3* | 99.68 | 99.62 | 99.62 | 96.44 | 99.72 |
| ViT-S/16 | 1.2 | 3.82 | 3.58 | 2.92 | **3.86** |
| ViT-B/16 | 0.70 | 2.58 | 2.46 | 1.32 | **2.54** |
| ViT-L/16 | 0.40 | **1.50** | 1.30 | 1.14 | 1.26 |
| Deit-T/16 | 2.25 | 6.64 | 6.50 | 5.22 | **6.94** |
| Deit-S/16 | 0.60 | 3.16 | 3.34 | 2.54 | **3.46** |
| Deit-B/16 | 0.73 | 2.62 | **2.85** | 2.12 | 2.70 |
| Swin-B/4/7 | 0.74 | 2.48 | **2.60** | 2.04 | 2.36 |
| SNN-VGG | 12.52 | 37.24 | 36.92 | 26.9 | **43.70** |
| SNN-ResNet | 32.54 | 59.68 | 57.57 | 49.72 | **62.26** |
| SNN-VGG/L | 13.30 | 38.74 | 38.86 | 29.04 | **46.56** |
| SNN-ResNet/L | 12.54 | 33.72 | 33.74 | 25.46 | **39.01** |
| DyNN-R50/96 | 3.64 | 8.08 | 7.90 | 6.68 | **8.64** |
| DyNN-R50/128 | 3.14 | 8.74 | 9.14 | 7.42 | **10.08** |
| DyNN-D121/96 | 3.82 | 7.28 | 7.54 | 6.40 | **8.14** |

