# OpenReview forum: "Provable Convergence of Clipped Normalized-gradient Heavy-Ball Momentum for Adversarial Attacks"
_ICLR.cc/2024/Conference — Submitted to ICLR 2024_

### Official Review · Reviewer_kjcR · 2023-10-21

**Soundness:** 2 fair
**Presentation:** 2 fair
**Contribution:** 2 fair
**Rating:** 5
**Confidence:** 3

**Summary:**

This paper proposes a method, clipped NGM, to improve the performance of adversarial attack. Intuition, theory, and numerical experiments are provided.

**Strengths:**

The paper is self-contained, with all intuition, theory, and numerical results included.

**Weaknesses:**

[1] A literature review is not sufficient. Checking through the reference list, the most recent literature is a paper from ICLR2021. The authors are encouraged to search for papers published in recent three years to highlight the difficulty of the problem and the novelty of their proposed method.

In addition, some literature is missing:

Croce, Francesco, et al. "Robustbench: a standardized adversarial robustness benchmark." arXiv preprint arXiv:2010.09670 (2020).

Croce, Francesco, and Matthias Hein. "Reliable evaluation of adversarial robustness with an ensemble of diverse parameter-free attacks." International conference on machine learning. PMLR, 2020.

Please compare the proposed method with the attack in the above literature. Robustbench is an important leaderboard evaluating the robustness of different training methods in adversarial training.

[2] When describing the results in Figure 3, it is mentioned that "the generated noise in each image is human-imperceptible". However, I think the noise in the proposed method is more obvious than the one from PGD, and such a difference leads to the difference in the attack success rates. Figure 3 is not convincing enough.

The authors may visualize the data under different attacks in their latent space to see whether any attack is easier to detect than others.

[3] The authors need to use different data sets in the experiments, e.g., CIFAR-10, CIFAR-100, ImageNet. The current experiments are not sufficient to demonstrate the improvements of the proposed method. The authors may also consider other neural network architectures, e.g., WideResNet models or ViT.

[4] My understanding of the proposed attack method is that it combines attack and some existing optimization methods. In addition, from how the proof is presented in the appendix, the proof follows some routine steps in proving optimization convergence, and the theoretical contribution is also limited. The authors may need to highlight the novelty and emphasize the difficulty or potential challenge when crafting their proposed method.

[5] The writing needs improvement. For example, the math notations can be improved. An attack algorithm aims to find a perturbation in x, and (9) and (10) are using x. However, in other formulas, the notation w is used. The notation w is often used for model parameters rather than the data x, confusing readers.

Another suggestion is to emphasize the purpose of each paragraph at the beginning of the paragraphs. When reading this paper, sometimes I have to read the whole paragraph, understand the details, and eventually get to know the purpose of the paragraph. Although the paragraphs still contain all the information, reading this paper is tiring.

**Questions:**

Please address my concerns in the weakness section.

---

> ### Author Response · Authors · 2023-11-16
> **Response to R4**
>
> Thank you for providing helpful comments. Due to space limitations, we only uploaded partial experimental results.
>
> Q1: New literature, other data-sets and neural network architectures.
>
> A1: Robustbench is indeed an important leaderboard evaluating the robustness of different training methods in adversarial training. In the revised version, we have cited the two given papers. We have searched for papers published in recent three years and find a closed-related survey, in which the adversarial transferability of AEs generated with representative attacks is re-evaluated across different types of DNNs, i.e., Wenqian Yu. et al. “Reliable Evaluation of Adversarial Transferability.” ArXiv abs/2306.08565 (2023).
>
> Since we only focus on evaluating the transferability of adversarial attack of gradient-based algorithms, we choose to compare our clipped NGM with all the mentioned gradient-based algorithms in Wenqian Yu. et al 2023 using their evaluation set, which includes PGD, MI-FGSM, NI-FGSM and VMI-FGSM. In fact, our computing power only allows us to do so. We use the same parameters ($\alpha$ and $\gamma$) decided by our previous CNN experiments. The experimental results are shown in the following tables, in which the setting of other gradient-based algorithms is directly from Wenqian Yu. et al 2023.
> |Attack|VGG16|Res50|Dense121|Inc-v3|Vit-S/16|Vit-B/16*|Vit-L/16|Deit-T/16|Deit-S/16|Deit-B/16|Swin-B/4/7|
> |-|-|-|-|-|-|-|-|-|-|-|-|
> |PGD|17.0|10.2|11.4|7.8|57.6|**100**|36.4|25.6|27.6|29.5|14.0|
> |MI|39.3|21.6|21.8|15.0|62.7|**100**|49.6|35.8|38.3|38.4|18.7|
> |NI|**40.5**|22.2|21.3|15.5|62.2|**100**|49.6|36.3|38.5|39.3|18.9|
> |VMI|28.2|14.8|14.3|9.5|47.2|99.5|33.0|25.2|26.6|24.9|10.4|
> |Ours|36.4|**25.1**|**25.8**|**19.4**|**77.2**|**100**|**64.6**|**44.9**|**50.2**|**51.5**|**20.1**|
>
> |Attack|VGG16*|Res50|Dense121|Inc-v3|Vit-S/16|Vit-B/16|Vit-L/16|Deit-T/16|Deit-S/16|Deit-B/16| Swin-B/4/7|
> |-|-|-|-|-|-|-|-|-|-|-|-|
> |PGD|**99.4**|19.8|16.7|7.5|1.7|1.1|0.5|4.8|1.6|1.2|2.0|
> |MI|99.3|41.0|39.0|23.0|5.3|3.5|1.5|12.9|6.2|3.2|4.8|
> |NI|99.2|41.8|39.7|22.8|5.7|3.3|1.4|12.3|5.6|3.4|4.6|
> |VMI|99.2|3.3|2.6|1.7|0.2|0.2|0.2|1.2|0.7|0.4|0.3|
> |Ours|**99.4**|**51.9**|**41.6**|**28.8**|**6.6**|**3.8**|**1.7**|**14.8**|**7.4**|**3.9**|**5.7**|
>
>
> We observe that our clipped NGM almost consistently outperforms MI-FGSM when using ViT-B/16 and VGG16 as the source model. Note that our algorithm is directly established upon MI-FGSM, we first focus on comparing with MI-FGSM by conducting the same experiments. The new experimental results further support our contribution.
>
> Due to the limitation of our computing power, it is impossible to conduct all the experiments you asked. Note for adversarial attacks, ILSVRC 2012 (a subset of ImageNet) is commonly used such as in (Dong et al., 2018; Lin et al., 2020; Wang et al., 2021; Wenqian Yu. et al., 2023). In the future, we will use Robustbench to evaluate robustness using more models and datasets. Please note that our purpose here is mainly to show the sign-operation can be replaced by the clipping strategies rather than the comprehensive performance evaluation against a wide range of attack methods and models. Thanks.
>
> Q2: Novelty, difficulty or potential challenge.
>
> A2: There is currently no convergence guarantee for a majority of gradient-based algorithms in adversarial attacks. While regular gradient and momentum methods do guarantee convergence, their practical performance is not satisfactory. To bridge this challenging theory-practice gap, we present a new algorithm called clipped NGM. It is inspired by the empirical success of sign-operation and normalized gradient in MI-FGSM rather than a simple combination of the known techniques. Another difficulty is that the step size conditions required to guarantee convergence of the regular clipped gradient algorithms are no longer satisfied due to using the normalized gradient. To overcome this difficulty, we present a new radius-varying clipping rule so that some routine steps in proving optimization convergence can be employed. It should be indicated that almost all optimization algorithms use some routine steps in convergence analysis.
>
> Q3: The noise human-imperceptible.
>
> A3: We have revised it to be “Whether the background is simple or complex, the difference between the noise images generated by MI-FGSM and our clipped NGM is almost human-imperceptible.”
> In NI-FGSM (Lin et al., ICLR2020), 12 randomly selected benign images and their corresponding adversarial images are visualized. They indicated that the generated noise in each image is human-imperceptible. Your suggestion will encourage us to do some future work about imperceptibility. Thanks.
>
> Q4: $ \boldsymbol{w} $ and $ \boldsymbol{x}$.
>
> A4: $ \boldsymbol{w} $ is for a pure optimization algorithm. $ \boldsymbol{x} $ is for describing an optimizer for adversarial attacks.
>
> Q5: The writing needs improvement.
>
> A5: Thanks. We will improve our presentation as soon as possible.

---

> ### Author Response · Authors · 2023-11-21
>
> We have conducted the comparisons, as shown in the following tables. The new experimental results still support our contribution.
>
> Transferability with VGG16 as the source model (T=10)
> |Attack|VGG16|Res50|Dense121|Inc-v3|Vit-S/16|Vit-B/16|Vit-L/16|Deit- T/16|Deit-S/16|Deit-B/16|Swin-B/4/7|SNN-VGG|SNN-Res|SNN-VGG/L|SNN-Res/L|DyNN-R50/96|DyNN-R50/128|DyNN-D121/96|
> |-|-|-|-|-|-|-|-|-|-|-|-|-|-|-|-|-|-|-|
> |PGD|**99.4**|19.8|16.7|7.5|1.7|1.1|0.5|4.8|1.6|1.2|2.0|93.5|47.9|92.6|27.4|5.4|5.9|5.6|
> |MI |99.3|41.0|39.0|23.0|5.3|3.5|1.5|12.9|6.2|3.2|4.8|98.2|73.3|98.0|57.3|10.8|14.5|11.3|
> |NI |99.2|41.8|39.7|22.8|5.7|3.3|1.4|12.3|5.6|3.4|4.6|98.2|72.8|98.1|57.3|11.4|14.5|11.4|
> |VMI |98.6|40.0|37.5|21.6|5.4|2.3|1.5|12.0|5.2|3.2|4.6|97.4|72.2|97.3|55.5|11.5|14.5|11.5|
> |Ours|**99.4**|**51.9**|**46.6**|**28.8**|**6.6**|**3.8**|**1.7**|**14.8**|**7.4**|**3.9**|**5.7**|**98.8**|**76.8**|**98.8**|**68.5**|**13.5**|**18.7**|**13.3**|
>
>
> Transferability with Inc-v3 as the source model (T=10)
> |Attack|VGG16|Res50|Dense121|Inc-v3|Vit-S/16|Vit-B/16|Vit-L/16|Deit- T/16|Deit-S/16|Deit-B/16|Swin-B/4/7|SNN-VGG|SNN-Res|SNN-VGG/L|SNN-Res/L|DyNN-R50/96|DyNN-R50/128|DyNN-D121/96|
> |-|-|-|-|-|-|-|-|-|-|-|-|-|-|-|-|-|-|-|
> |PGD|9.4|6.0|5.6|**99.7**|1.2|0.7|0.4|2.3|0.6|0.7|0.7|12.5|32.5|13.3|12.5|3.6|3.1|3.8|
> |MI |25.4|17.3|16.4|99.6|3.8|**2.6**|**1.5**|6.6|3.2|2.6|2.5|37.2|59.7|38.7|33.7|8.1|8.7|7.3|
> |NI |24.9|16.8|16.4|99.6|3.6|2.5|1.3|6.5|3.3|**2.9**|**2.6**|36.9|57.5|38.9|33.7|7.9|9.1|7.5|
> |VMI |6.5|1.9|1.6|67.8|0.2|0.2|0.2|0.9|0.1|0.1|0.1|20.1|51.0|20.5|20.0|2.9|1.6|2.7|
> |Ours |**33.4**|**21.1**|**18.5**|**99.7**|**3.9**|2.5|1.3|**6.9**|**3.5**|2.7|2.4|**43.7**|**62.3**|**46.6**|**39.0**|**8.6**|**10.1**|**8.1**|
>
> Transferability with ViT-B/16 as the source model (T=10)
> |Attack|VGG16|Res50|Dense121|Inc-v3|Vit-S/16|Vit-B/16|Vit-L/16|Deit- T/16|Deit-S/16|Deit-B/16|Swin-B/4/7|SNN-VGG|SNN-Res|SNN-VGG/L|SNN-Res/L|DyNN-R50/96|DyNN-R50/128|DyNN-D121/96|
> |-|-|-|-|-|-|-|-|-|-|-|-|-|-|-|-|-|-|-|
> |PGD|29.6|20.6|20.4|15.1|59.8|**100**|45.3|43.3|48.3|50.4|27.8|48.9|72.6|53.5|55.1|13.4|15.5|13.9|
> |MI|29.7|20.4|21.2|15.9|61.0|**100**|44.4|42.4|48.1|49.4|27.1|49.0|72.2|54.1|52.9|12.5|14.6|13.6|
> |NI|29.8|20.8|20.9|16.1|60.8|**100**|45.8|43.0|49.0|51.0|27.8|50.4|**74.6**|55.5|55.3|13.0|15.4|14.0|
> |VMI|7.4|1.8|1.6|1.3|24.6|26.2|0.4|3.2|1.1|0.4|0.1|22.8|59.7|25.8|26.1|3.4|2.4|3.2|
> |Ours|36.4|**25.0**|**25.0**|**18.6**|**65.0**|**100**|**64.4**|**44.2**|**50.2**|**51.4**|**29.9**|**55.6**|74.5|**61.6**|**61.3**|**15.6**|**18.7**|**17.1**|

---

> > ### Comment · Reviewer_kjcR · 2023-12-04
> >
> > I appreciate the authors in providing more experiments and commenting on my concerns. I raised my score from 3 to 5. I'm still concerned about the performance of the proposed algorithm. As mentioned in my original review, I think the noise in the proposed method is more obvious than the one from PGD, and such a difference leads to the difference in the attack success rates.

---

### Official Review · Reviewer_mvi6 · 2023-11-01

**Soundness:** 3 good
**Presentation:** 3 good
**Contribution:** 2 fair
**Rating:** 5
**Confidence:** 3

**Summary:**

The paper proposes a new adversarial attack algorithm based on gradient clipping, momentum, and normalized gradient. The algorithm is proven to converge and experiments show it has superior performance than existing attack algorithms.

**Strengths:**

1. The paper extends gradient clipping into the setting of adversarial attack which is a novel contribution.
2. Theoretical analysis is provided for the proposed algorithm to show it is convergent.

**Weaknesses:**

1. The paper mentioned a few times that signsgd is non-convergent, but did not provide any example of its non-convergent. The claim should be consolidated and more importantly, for the adversarial attack problem. My intuition tells me signsgd might be convergent for adversarial attack problems with box (l1) constraint, it may diverge for l2 constraints.
2. Gradient clipping is shown to accelerate training theoretically in prior works, I missed any solid acceleration discussions in the paper.

**Questions:**

Could the authors provide some clarifications to the two weaknesses?

---

> ### Author Response · Authors · 2023-11-16
> **Response to R3**
>
> Thank you for your efforts reviewing our paper and providing helpful comments and suggestions. We are looking forward to your response.
>
> Q1: The non-convergence of SignSGD.
>
> A1: In Karimireddy et al., 2019 (page 4 and 5), three simple convex counter-examples show that signSGD does not converge to the optimum.
>
> Q2: The solid acceleration discussions of gradient clipping.
>
> A2: In fact, the acceleration of gradient clipping needs some assumptions such as the smoothness condition in Zhang et al., 2020a. For general convex functions, as far as we know, it remains unknown whether clipping can accelerate SGD. In this paper, we only focus on proving that the clipped NGM can attain the optimal averaging convergence, as opposed to the non-convergence of signSGD. Please note that our main purpose is to show that the clipping technique can serve as an alternative to the dominant sign operation in adversarial attacks.
>
> Of course, your question is very interesting. Our intuition is that the clipped NGM should have a better bound with smaller factors than SGD due to the rescaling of gradients. Your suggestion inspires us to carefully analyze the derived bound of Theorem 3.4 in our future work. At least, we can discuss the bounds in some special cases (e.g., we can let the two parts in the derived bounds be equal and discuss their minimal value). More generally, we can discuss whether we can use the lower bound from the clipped gradients to take the place of the original bound of gradients in the proof. Thanks.

---

### Official Review · Reviewer_nPVd · 2023-11-02

**Soundness:** 3 good
**Presentation:** 2 fair
**Contribution:** 2 fair
**Rating:** 5
**Confidence:** 2

**Summary:**

The authors propose a clipped momentum method (clipped NGM), in which the normalized-gradient momentum (NGM) is clipped as its update direction with a radius-varying clipping rule.  They show that the clipped NGM attains its averaging convergence for general
constrained convex problems. Numerical experiments are given to demonstrate the efficiency of the clipped NGM.

**Strengths:**

Numerical experiments and comparisons with the sign-momentum MI-FGSM are given in some adversarial attack problems.

**Weaknesses:**

The basic idea of the proposed algorithm, clipped NGM, looks a bit unnatural to me. The authors state that they propose the algorithm based on replacing the sign operator in the sign-momentum algorithm-MIFGSM by a clipping operator. As we all know, in the sign-momentum-based algorithm, using the sign operator, the normalization of the gradient in the momentum update is natural as one would lose the magnitude information once taking the sign operator. For the clipped NGM, it looks like the normalization of the gradient could be unnecessary, as one already has another energy-clipping operator in the iterative updates.  Without considering the normalization gradient step in the momentum updates, the algorithm looks similar to the one studied in (Mai & Johansson, 2021). From this point of view, numerical comparisons with the clipped momentum algorithm by  (Mai & Johansson, 2021) should be better given.

The presentation could be further improved. The introduction of the motivation and the algorithms looks like a composition of pieces of different algorithms in different areas.

In the introduction of the background of some related algorithms, the authors mention Adam algorithm and say that the drawback of Adam is its non-convergence issue. However, to my best knowledge, Adam indeed can converge in some parameter settings. Besides, Adam is one of the popular algorithms in ML area. However, the authors do not provide any numerical comparisions with Adam algorithm.

The derived theoretical results are for the convex constraints problems with bounded gradient and domain assumptions, which is a bit restricted in a world filled with non-convex optimization problems.

**Questions:**

see the above

---

> ### Author Response · Authors · 2023-11-16
> **Response to R2**
>
> Thank you for your efforts reviewing our paper and providing helpful comments and suggestions. We are looking forward to your response.
>
> Q1: Clipped NGM looks a bit unnatural.
>
> A1: Yes. We also have this feeling when reading papers about MI-FGSM and NI-FGSM, in which the sign operation and normalizing gradient are simultaneously used. We have done a lot of experiments and find that normalizing the gradient with its norm is a key to the success of many gradient methods. On the other hand, the experimental results will be far from satisfactory when we directly employ the regular (natural) algorithms such as SGD, HB, NAG, Adam and clipped-like algorithms. In a word, many gradient-based optimizers in adversarial attacks look empirical but have practical performance while our clipped NGM is a new algorithm that has both convergence guarantee and practical performance.
>
> Q2: Numerical comparisons with Adam and the clipped-M by (Mai & Johansson, 2021).
>
> A2: We have conducted the comparisons, as shown in the following table. It is not surprising at all that the performance of Adam and Clipped-M is even worse than MI-FGSM. One cause for such failures is that the normalized gradient is not used in these algorithms.
>
> **Model: Inc-V3**
> | Attack | Inc-v3 | Inc-v4 | IncRes-v2 | Res-101 | v3ens3 | v3ens4 | v2ens | v3Adv |
> | ---- | ---- | ---- | ---- | ---- | ---- | ---- | ---- | ---- |
> | Adam | 99.5 | 38.6 | 35.6 | 29.1 | 12.9 | 11.4 | 6.2 | 16.3 |
> | MI-FGSM | **100** | 44.3 | 42.1 | 36.1 | 14.3 | 13.7 | 6.2 | 18.9 |
> | Clipped-M | 96.5 | 42.9 | 40.6 | 34.1 | 10.9 | 10.1 | 5.6 | 17.7 |
> | Clipped-NGM | **100** | **48.8** | **45.9** | **38.9** | **16.2** | **14.6** | **7.1** | **23.9** |
>
> **Model: Inc-v4**
> | Attack | Inc-v3 | Inc-v4 | IncRes-v2 | Res-101 | v3ens3 | v3ens4 | v2ens | v3Adv |
> | ---- | ---- | ---- | ---- | ---- | ---- | ---- | ---- | ---- |
> | Adam | 48.7 | 99.6 | 39.8 | 35.5 | 16.0 | 13.7 | 7.5 | 15.1 |
> | MI-FGSM | 56.8 | 99.7 | 46.4 | **42.5** | 16.6 | 14.7 | 8.0 | 18.4 |
> | Clipped-M | 51.3 | 90.2 | 43.2 | 40.4 | 15.1 | 13.6 | 6.8 | 20.0 |
> | Clipped-NGM | **57.3** | **99.9** | **47.8** | 42.1 | **19.2** | **15.0** | **7.6** | **21.5** |
>
> **Model: IncRes-v2**
> | Attack | Inc-v3 | Inc-v4 | IncRes-v2 | Res-101 | v3ens3 | v3ens4 | v2ens | v3Adv |
> | ---- | ---- | ---- | ---- | ---- | ---- | ---- | ---- | ---- |
> | Adam | 54.8 | 46.1 | 98.2 | 37.7 | 21.2 | 16.0 | 11.8 | 20.1 |
> | MI-FGSM | 60.0 | 50.6 | 98.0 | **44.1** | 21.6 | 16.4 | 11.3 | 23.0 |
> | Clipped-M | 47.4 | 41.9 | 79.2 | 35.1 | 22.9 | 16.4 | 13.3 | 23.4 |
> | Clipped-NGM | **60.0** | **52.9** | **99.0** | 44.0 | **24.3** | **18.8** | **13.7** | **29.0** |
>
> **Model: Res101**
> | Attack | Inc-v3 | Inc-v4 | IncRes-v2 | Res-101 | v3ens3 | v3ens4 | v2ens | v3Adv |
> | ---- | ---- | ---- | ---- | ---- | ---- | ---- | ---- | ---- |
> | Adam | 48.6 | 43.7 | 41.4 | 97.3 | 23.2 | 20.4 | 12.4 | 21.2 |
> | MI-FGSM | 56.6 | 51.6 | 48.7 | **99.3** | 24.1 | 22.0 | 12.1 | 24.9 |
> | Clipped-M | 50.0 | 45.1 | 45.1 | 87.5 | 21.4 | 17.5 | 10.9 | 24.9 |
> | Clipped-NGM | **59.6** | **53.5** | **51.4** | **99.3** | **29.3** | **24.4** | **16.4** | **34.4** |
>
> Q3: The convergence of Adam.
>
> A3: In Reddi et al., 2018 (page 4), they provided an explicit example of a simple convex optimization setting where Adam does not converge to the optimal solution.
>
> Q4: The convex constraints problems with bounded gradient and domain assumptions.
>
> A4: The bounded gradient and domain assumptions are easily satisfied for adversarial attack optimization problem due to the human-imperceptible constraint. Note that $J(\boldsymbol{x}^{adv}, y) > J(\boldsymbol{x}, y)$ when an adversarial example can be successfully generated from a correctly classified sample $\boldsymbol{x}$. Intuitively, we can assume that the objective function is locally concave. Since the constrained domain is generally small due to the human-imperceptible constraint, we can further assume the objective function to be concave without loss of generality.
>
> Q5: A bit restricted in a world filled with non-convex optimization problems.
>
> A5: Although the nonconvex problems are recently considered by many papers, their convergence analysis still needs many additional assumptions (such as smoothness) that actually does not hold in many deep learning models (such as ResNet using the activation function ReLU). In fact, convexity is regarded as a mild and basic assumption in the optimization community. In deep learning tasks, it does not prevent the convex optimization scheme (e.g. Adam) from motivating great empirical successes.
>
> Q6: The presentation.
>
> A6: Thanks. We will improve our presentation as soon as possible.

---

### Official Review · Reviewer_F2ht · 2023-11-04

**Soundness:** 2 fair
**Presentation:** 3 good
**Contribution:** 2 fair
**Rating:** 5
**Confidence:** 3

**Summary:**

This paper proposes a clipped normalized-gradient heavy-ball momentum method for generating adversarial examples. Moreover, the authors also provide the convergence analysis of the proposed algorithm. Some experimental results show the performance of the proposed method.

**Strengths:**

1. This paper proposes a clipped normalized-gradient heavy-ball momentum method for generating adversarial examples.
2. Moreover, the authors also provide the convergence analysis of the proposed algorithm.
3. Some experimental results show the performance of the proposed method.

**Weaknesses:**

Although the paper is theoretically and experimental sound, there are still some questions need to be discussed in this paper:
1.	The novelty of this paper is limited. In other words, the main techniques used in this paper include one clipped technique and a heavy-ball momentum acceleration method. However, all the techniques are widely used in previous works.
2.	The convergence analysis of the proposed algorithm is provided for the constrained convex problem. But in fact, the problem of generating adversarial examples is non-convex. Therefore, the authors should provide the convergence analysis of the proposed algorithm for solving both convex and non-convex optimization problems.
3.	Eq. (15) may be questionable. Note that the clipping operator in Eq. (8) should be element-wise, and thus Eq. (15) includes the element-wise product of two vectors.
4.	The experimental results are not convincing. The authors should compare the proposed algorithm with more recently proposed algorithms.
5.	Both the English language and equations in this paper need to be improved. For instance, what’s the definition of the (L_0, L-1)-smooth function?

**Questions:**

Although the paper is theoretically and experimental sound, there are still some questions need to be discussed in this paper:
1.	The novelty of this paper is limited. In other words, the main techniques used in this paper include one clipped technique and a heavy-ball momentum acceleration method. However, all the techniques are widely used in previous works.
2.	The convergence analysis of the proposed algorithm is provided for the constrained convex problem. But in fact, the problem of generating adversarial examples is non-convex. Therefore, the authors should provide the convergence analysis of the proposed algorithm for solving both convex and non-convex optimization problems.
3.	Eq. (15) may be questionable. Note that the clipping operator in Eq. (8) should be element-wise, and thus Eq. (15) includes the element-wise product of two vectors.
4.	The experimental results are not convincing. The authors should compare the proposed algorithm with more recently proposed algorithms.
5.	Both the English language and equations in this paper need to be improved. For instance, what’s the definition of the (L_0, L-1)-smooth function?

---

> ### Author Response · Authors · 2023-11-16
> **Response to R1**
>
> Thank you for your valuable feedback.
>
> Q1: All the techniques are widely used in previous works.
>
> A1: Indeed, gradient clipping is a rescaling method in optimization theory and the normalized gradient technique is from MI-FGSM. However, some difficulties in convergence are caused when the gradient is normalized with its norm. This may explain why there is a lack of convergence analysis for MI-FGSM and NI-FGSM so far. To overcome the non-convergence of sign-like methods, we first present a clipped NGM, which is motivated by the empirical success of sign-operation and normalized gradient in MI-FGSM. Obviously, it is not a simple combination of the known techniques.
>
> It is a well-known fact that the convergence of each optimization algorithm heavily depends on its step-size. To guarantee the convergence of our clipped NGM, a new step-size rule is proposed. Theoretically and empirically, we find that the clipping technique can serve as an alternative to the dominant sign operation in adversarial attacks, which is an interesting point in this paper.
>
> Q2: The problem of generating AEs is non-convex.
>
> A2: First of all, note that $J(\boldsymbol{x}^{adv}, y) > J(\boldsymbol{x}, y)$ when an adversarial example can be successfully generated from a correctly classified sample $\boldsymbol{x}$. Intuitively, we now can assume that the objective function is locally concave. Since the constrained domain is generally small due to the human-imperceptible constraint, we can further assume the objective function to be concave without loss of generality.
>
> On the other hand, the convergence of almost all the optimizers including the well-known SGD and Adam also need assumptions. In fact, convexity is regarded as a mild and basic assumption in the optimization community. In deep learning tasks, it does not prevent the convex optimization scheme (e.g. Adam) from motivating great empirical successes. It is impossible to get convergence without any assumption. Besides, although the nonconvex problems have been recently considered by many papers, their convergence analysis still needs many additional assumptions (such as smoothness) that actually does not hold in many deep learning models (such as ResNet using the activation function ReLU).
>
> Q3: Eq. (15) may be questionable.
>
> A3: It is not questionable. Eq. (15) is similar to Eq. (7) but using the normalized-gradient like that in MI-FGSM. We apologize for the original formulation $\text{clip}_{\gamma}: \mathbb{R}^{N} \rightarrow\mathbb{R}^{N}$.
>
> It should be $ \text{clip}_{\gamma}: \mathbb{R}^{N} \rightarrow\mathbb{R}  $.
>
>
> Q4: Comparing the proposed algorithm with more recently proposed algorithms.
>
> A4: Our purpose here is mainly to show the sign-operation can be replaced by the clipping strategies rather than a comprehensive performance evaluation against a wide range of attack methods and models. Naturally, since our algorithm is directly established upon MI-FGSM, we mainly focus on comparing with MI-FGSM by conducting the same experiments. Comparison with the more recent gradient-based method VMI-FGSM (Wang et al., CVPR2021) can be found in the Appendix. Besides, the transferability comparison with gradient-based algorithms across different network architectures is added in this rebuttal (Q1 of R4). The new experimental results further support our contribution to this paper.
>
> Q5: The definition of the (L_0, L-1)-smooth function
>
> A5: It can be found in Zhang et al., NeurIPS 2020 (page 2, Definition 1.1).

---

### Author Response · Authors · 2023-11-20

We would like to thank all reviewers for their constructive and insightful comments. We have just finished our experiments and uploaded a revised version. The main change is that more experiments are added in the Conclusion section and Appendix to address your concerns. We will update this paper again in a few days with further revisions (e.g. improving presentation).

---

> ### Author Response · Authors · 2023-11-23
>
> Dear Reviewers,
>
> We have finished comparing our clipped NGM with all the mentioned gradient-based algorithms and models in Wenqian Yu. et al. using their evaluation set. The new experimental results still  support our contribution.
>
> We have  uploaded a revised version and we are looking forward to your response. Thanks.
>
> Wenqian Yu. et al. “Reliable Evaluation of Adversarial Transferability.” ArXiv abs/2306.08565 (2023).

---

### Meta-Review · Area_Chair_wQYq · 2023-12-04

**Metareview:**

This paper propose a new optimization technique for Adversarial attacks based on clipping and normalized gradient. The authors show that the method converge in the convex case and evaluate empirically their method on CIFAR10 models.
The strenght of the paper is that it is well motivated theoretically (The method can converge in the convex case unlike sign gradient method).
However, there are no investigation that this problem (the baselines cannot converge) is actually occurring in practice (The baselines may actually be converging in practice because of the specific structure of the problem). Thus there is a gap between the motivation and the practice that could be closed. I suggest the authors to put some effort in that direction.

In my opinion, the main weakness of this paper is the experimental evaluation. It is not 100% that the comparison with the baseline have been done in a fair way.
In particular the author should consider:
- standard benchmark code
- Tune hyperparameters for other techniques. (alpha = \epsilon/T is not the best step size for most methods) You use alpha = 18 while the baseline use alpha = 1.6. It seems unfair for the baseline.
- Other datasets than CIFAR10

**Justification For Why Not Higher Score:**

There was a consensus toward that a better experimental evaluation was needed.
I read the paper and I agree with that point. More especially for adversarial examples, the experiments needs to be bulletproof as there are many way to under-evaluate the baseline.

**Justification For Why Not Lower Score:**

N\A

---

### Decision · Program_Chairs · 2024-01-16

Reject